# Can telehealth expansion boost health care utilization specifically for patients with substance use disorders relative to patients with other types of chronic disease?

Alyssa Shell Tilhou[1]*, Laura Dague[2], Preeti Chachlani[3], Marguerite Burns[4]

**1** Department of Family Medicine, Boston University School of Medicine/Boston Medical Center, Boston, MA, United States of America, **2** Public Service & Administration, Texas A&M University, College Station, TX, United States of America, **3** Institute for Research on Poverty, University of Wisconsin—Madison, Madison, WI, United States of America, **4** Department of Population Health Sciences, University of Wisconsin—Madison, Madison, United States of America

* alyssatilhoumd@gmail.com, alyssa.tilhou@bmc.org

## Abstract

### Objective

Patients with substance use disorders (SUDs) exhibit low healthcare utilization despite high risk of poor outcomes. Telehealth expansion may boost utilization, but it is unclear whether telehealth can increase utilization for patients with SUDs beyond that expected for other chronic diseases amenable to remote treatment, like type 2 diabetes. This information is needed by health systems striving to improve SUD outcomes, specifically. This study compared the impact of telehealth expansion during the COVID-19 public health emergency (PHE) on utilization for patients with SUDs and diabetes.

### Methods

Using Wisconsin Medicaid administrative, enrollment and claims data 12/1/2018-12/31/2020, this cohort study included nonpregnant, nondisabled adults 19–64 years with SUDs (N = 17,336) or diabetes (N = 8,499). Outcomes included having a primary care visit in the week (any, and telehealth) for any diagnosis, or a SUD or diabetes diagnosis; and the weekly fraction of visits completed by telehealth. Logistic and fractional regression examined outcomes pre- and post-PHE. Covariates included age, sex, race, ethnicity, income, geography, and comorbid medical and psychotic disorders.

### Results

Post-PHE, patients with SUDs exhibited greater likelihood of telehealth utilization (percentage point difference (PPD) per person-week: 0.2; 95% CI: 0.001–0.003; p<0.001) and greater fractional telehealth use (PPD: 1.8; 95%CI: 0.002–0.033; p = 0.025) than patients with diabetes despite a larger overall drop in visits (PPD: -0.5; 95%CI: -0.007- -0.003; p<0.001).

**Data Availability Statement:** Data Availability: The data for this paper are not publicly available. The paper uses linkages of proprietary data from the Wisconsin Department of Health Services housed

at the Institute for Research on Poverty at the University of Wisconsin. The construction of and access to these data are governed by data sharing agreements between the University of Wisconsin and the agency that prohibit any redisclosure of the data. Sharing of individual, row-level data is strictly prohibited by each of these existing data use agreements. This prohibition extends to derived datasets based on the raw claims data that aggregate claims, for example, at the person- or person-year level and all data sets to which they are linked at the person-level. Others interested in using the data may inquire through IRP at https://www.irp.wisc.edu/wadc/ and apply for permission. Contact: Tim Connor. tim.connor@wisc.edu.

**Funding:** Funding for this research comes from the Wisconsin Department of Health Services through an 1115 waiver evaluation received by MB and LD. Dr. Alyssa Tilhou is funded by National Institute on Drug Abuse K08DA058052. The funder did not play a role in study design, data collection and analysis, decision to publish, or preparation of the manuscript.

**Competing interests:** The authors have declared no completing interests exist.

## Conclusions

Following telehealth expansion, patients with SUDs exhibited greater likelihood of telehealth utilization than patients with diabetes. This advantage lessened the substantial PHE-induced healthcare disruption experienced by patients with SUDs. Telehealth may boost utilization for patients with SUDs.

## Introduction

Patients with substance use disorders (SUDs) comprise a particularly vulnerable patient population due to their high risk for poor health outcomes alongside low health care utilization: SUDs are associated with the development of heart, liver and lung diseases, cancers, infectious diseases, and an assortment of mental health disorders such as depression, anxiety and suicidal ideation [1, 2]. Yet, patients with SUDs exhibit low rates of treatment utilization for both medical [3] and addictive disorders [4, 5]. Moreover, the majority of people living with SUDs never utilize addiction services [4] and frequently receive suboptimal medical care [6]. As a result, improving access to medical and substance use treatment services is central to improving outcomes for this patient population.

Telehealth expansion may be one strategy to enhance health care utilization for patients with SUDs by reducing barriers to care, thereby creating an opportunity to reduce health disparities for this patient population. Common barriers to care for this population include affordability and availability of services, stigma, and prevalence of comorbid mental health disorders [7, 8]. Telehealth can help overcome these barriers by simplifying treatment logistics, enhancing patient anonymity, and reducing the costs of seeking care [9–12]. Telehealth may also represent an effective modality for treating patients with SUDs given mounting evidence that the core of SUD care–counseling and medication management–can be delivered via telehealth with at or near comparable outcomes to in-person care [13–16]. Yet, by requiring technology, internet and remote privacy, telehealth expansion could simultaneously deepen inequities for individuals with financial insecurity and homelessness, common among patients with SUDs [12, 17, 18]. Given these competing forces, it is unclear whether telehealth expansion can increase health care utilization for this patient population. More evidence is needed for health systems considering how and whether to deepen their investment in telehealth capabilities for patients with SUDs.

Comparing telehealth uptake across different patient populations could demonstrate whether telehealth expansion offers a particular advantage for patients with SUDs. An appropriate comparison group would exhibit similar insurance coverage and socioeconomic need, duration of treatment, treatment settings used, and clinical appropriateness for remote care. Low-income patients with chronic medical disease frequently have high medical need and require ongoing evaluation like patients with SUDs [11, 19]. Specifically, patients with type 2 diabetes offer a compelling comparison group. In contrast with management of cardiac, pulmonary, renal and hepatic disease, which is more substantially improved by in-person examination and laboratory data, much of the management of type 2 diabetes can occur remotely via collaborative retrospective evaluation of home glucose monitoring [20]. The impact of addiction on psychosocial functioning, including those behaviors involved in seeking and utilizing care, distinguishes patients with SUDs from those with diabetes [18]. As a result, examining utilization among patients with SUDs relative to patients with diabetes may reveal whether telehealth can increase treatment utilization specifically for patients with SUDs.

The rapid telehealth expansion prompted by the COVID-19 public health emergency (PHE) offers an opportunity to examine differences in telehealth uptake for patients with SUDs relative to diabetes [21, 22]. While studies have reported on trends in telehealth use during the pandemic for patients with SUDs, comparative analyses remain rare. In addition, while studies have examined telehealth expansion in populations with Medicare and private insurance [21–24], fewer have described trends for low-income patient populations such as those with Medicaid [22]. Finally, while studies have evaluated telehealth incorporation into subspecialty practice [21], mental health care [25], emergency room services [26], and hospital medicine [27], less work has focused on primary care [21]. Given the central role of primary care in providing addiction and medical services to patients with SUDs and diabetes [28–31], examining the impact of telehealth incorporation on primary care utilization holds particular relevance for reducing utilization disparities for patients with SUDs.

This study aimed to measure changes in utilization of in-person and telehealth primary care services for adult Wisconsin Medicaid beneficiaries with SUDs during the rapid telehealth expansion period prompted by the COVID-19 PHE. To understand the degree to which changes in utilization are different for SUDs, we compare trends in treatment utilization for patients with SUDs with those from a cohort of Wisconsin Medicaid beneficiaries with type 2 diabetes.

## Methods

### Data source

Study data are from Wisconsin Medicaid administrative, enrollment and claims data from December 2018 through December 2020. Claims were used to identify beneficiaries with an SUD or type 2 diabetes (herein, "diabetes") diagnosis, as well as to identify the presence of a comorbid psychotic diagnosis. Enrollment data was used to obtain demographic characteristics of the two cohorts at baseline (June 2019) including eligibility category, age, sex, race, ethnicity, education, income and geography. In these data, race and ethnicity are usually obtained via self-identification but sometimes are reported by caseworkers.

### Population

The study cohorts included nonpregnant, nondisabled, noninstitutionalized adults ages 19 to 64 years eligible for Wisconsin Medicaid as either parents/caretakers or childless adults (CLA). Individuals who turned 65 during the study period were excluded. We required continuous enrollment from June 2019 through December 2020. We elected to require continuous enrollment to 1) better isolate the effects of telehealth expansion on utilization and b) minimize bias due to compositional changes. Maintenance of eligibility (MOE) protections provided by the Families First Coronavirus Response Act prohibited Medicaid beneficiary disenrollment during the PHE and so more beneficiaries than typical were continuously enrolled [32]. Continuous enrollment, defined as an enrollment gap of no more than 1 month, identified 143,992 individuals. Among those continuously enrolled during the study period, we identified the SUD and diabetes cohorts as those with at least 1 claim with an SUD or diabetes diagnosis in the six months before the study period (December 2018 –May 2019) in the outpatient, inpatient or emergency department setting. Continuous enrollment was not required in this six-month prior period. Patients with both an SUD and diabetes diagnosis were retained in the SUD cohort only. Diagnoses were identified using the International Statistical Classification of Diseases and Related Health Problems, Tenth Revision (ICD-10) codes. For SUD diagnoses, we included ICD-10 codes for alcohol, opioid, cannabis, sedative, stimulant, and other psychoactive substance use disorders (F10-F19). We excluded nicotine (F17) and miscellaneous

SUDs (F550-F558: antacids, herbal remedies, laxatives, steroids, vitamins and other non-psychoactive substances). For diabetes, we included ICD-10 codes E1100-E118 and E119. See S1 and S2 Appendices for additional detail regarding the underlying population and cohort construction.

## Outcome assessment and covariates

We defined the following outcomes: having a primary care office visit in the week (any modality and telehealth); and having a primary care office visit in the week for a SUD (SUD cohort) or diabetes (diabetes cohort) diagnosis (any modality and telehealth). We identified primary care visits using a combination of provider specialty codes and rendering provider taxonomy to identify visits completed by a primary care provider and/or in a primary care location. We identified telehealth based on the procedure code, or the presence of either a place of service code or modifier indicating telehealth consistent with state guidelines for providers. Visits that addressed diabetes or a SUD, specifically, were identified via diagnosis claims in association with the office visit. For additional details see S3 Appendix. Covariates included age, sex, race, ethnicity, income (as percentage of the federal poverty limit (FPL)), geography (residing in a rural or urban county or missing), presence of a comorbid chronic medical condition and presence of a comorbid psychotic disorder. In most cases, race and ethnicity data are obtained via self-identification, though occasionally reported by caseworkers. The presence of comorbid chronic medical or psychotic disorders was identified in the six months prior to study start via ICD-10 codes. Chronic medical conditions included asthma, chronic kidney disease, chronic obstructive pulmonary disease, coronary artery disease, hypertension, thyroid disorders, heart failure, chronic liver disease, and osteoarthritis [33]. Chronic psychotic disorders included schizophrenia, bipolar disorder, other psychotic disorders and diagnoses with psychotic features. The pre and post-PHE periods were defined as 6/1/2019-3/13/2020 and 3/14/2020-12/31/2020, respectively.

## Statistical analysis

Baseline characteristics were summarized and differences between the SUD and diabetes cohorts were evaluated using chi-square tests for factors with more than one level and two-sided t-tests. Visit rates were estimated as proportion of the cohort with a visit at the person-week level (any modality and by telehealth, specifically). Analyses were repeated for disease-specific visits. We conducted logistic regression to test for differences between the SUD and diabetes cohorts in: 1) the change in probability of having a primary care visit in the week (any modality, any diagnosis) post-PHE (Model 1), and 2) the change in probability of having a primary care visit by telehealth (any diagnosis) post-PHE (Model 2). We conducted fractional regression to test for differences in the fraction of primary care visits that incorporated telehealth between the SUD and diabetes cohorts in the post period and clustered at the beneficiary level (Model 3). All models adjusted for key characteristics including age, sex, race, ethnicity, FPL, geography, presence of a comorbid chronic medical condition and presence of a psychotic disorder. We excluded education due to conceptual overlap with income. We present results as predicted risk and marginal risk differences [34]. Please see S4 Appendix for technical details. Analyses were conducted using Stata statistical software (version 17.0; Stata Corp, LLP). The statistical significance level was set at 0.05. Analyses were conducted August 2021 to August 2022.

This study was determined exempt from review and informed consent by the University of Wisconsin's institutional review board (common rule, category 5). The authors did not have access to information that could identify individual participants during or after data collection.

This study followed the Strengthening the Reporting of Observational Studies in Epidemiology (STROBE) guidelines.

## Results

We identified 143,992 individuals who met the age, eligibility, and enrollment continuity criteria. Of these, 17,336 and 8,499 individuals met criteria for entry to the SUD and diabetes cohorts, respectively (Table 1). Beneficiaries with SUDs were more often white (71.13% vs 57.42%; p<0.001), reported income below 50% FPL (83.99% vs 66.58%; p<0.001), and

**Table 1. Characteristics of Wisconsin Medicaid beneficiaries with either a substance use disorder or diabetes.**

| | SUD Cohort | | Diabetes Cohort | | P value |
|---|---|---|---|---|---|
| N = unique subjects | 17,336 | 100.00% | 8,499 | 100.00% | <0.001 |
| **Comorbid diagnoses** | | | | | |
| Psychotic disorder | 2,680 | 15.46% | 452 | 5.32% | <0.001 |
| Chronic medical condition | 5,845 | 33.72% | 5,748 | 67.63% | <0.001 |
| **Sex** | | | | | |
| Female | 8,301 | 47.88% | 4,703 | 55.34% | <0.001 |
| Male | 9,035 | 52.12% | 3,796 | 44.66% | |
| **Age** | | | | | |
| Mean age (SD) | 38.88(10.34) | | 48(9.88) | | <0.001 |
| **Race** | | | | | |
| American Indian | 799 | 4.62% | 300 | 3.53% | <0.001 |
| Asian | 107 | 0.62% | 424 | 4.99% | |
| Black | 2,525 | 14.54% | 1,581 | 18.60% | |
| Multiracial | 368 | 2.09% | 126 | 1.48% | |
| Pacific Islander | 15 | 0.09% | 23 | 0.27% | |
| White | 12,340 | 71.13% | 4,880 | 57.42% | |
| Race Missing | 1,182 | 6.91% | 1,165 | 13.71% | |
| **Ethnicity** | | | | | |
| Hispanic | 1,154 | 6.63% | 1,015 | 11.94% | <0.001 |
| Not Hispanic | 15,972 | 92.13% | 7,334 | 86.29% | |
| Missing | 210 | 1.24% | 150 | 1.76% | |
| **Education** | | | | | |
| More than High School | 10,255 | 59.12% | 4,452 | 52.38% | <0.001 |
| Less than High School | 3,440 | 19.85% | 1,558 | 18.33% | |
| Missing | 3,641 | 21.03% | 2,489 | 29.29% | |
| **Income** | | | | | |
| ≤50% FPL | 14,560 | 83.99% | 5,659 | 66.58% | <0.001 |
| 50–100% FPL | 2,776 | 16.01% | 2,840 | 33.42% | |
| >100% FPL | 0 | 0.00% | 0 | 0.00% | |
| **Geography** | | | | | |
| Urban | 11,223 | 64.66% | 5,288 | 62.22% | <0.001 |
| Rural | 3,573 | 20.59% | 1,885 | 22.18% | |
| Missing | 2,540 | 14.74% | 1,326 | 15.60% | |
| **Cohort percentage with ≥1 visit in the week** | | | | | |
| Pre-PHE (All primary care visit types) | 7.90% | | 7.10% | | |
| Post-PHE (All primary care visit types) | 6.13% | | 5.85% | | |
| Pre-PHE (Primary care telehealth only) | 0.00% | | 0.00% | | |
| Post-PHE (Primary care telehealth only) | 1.48% | | 1.34% | | |

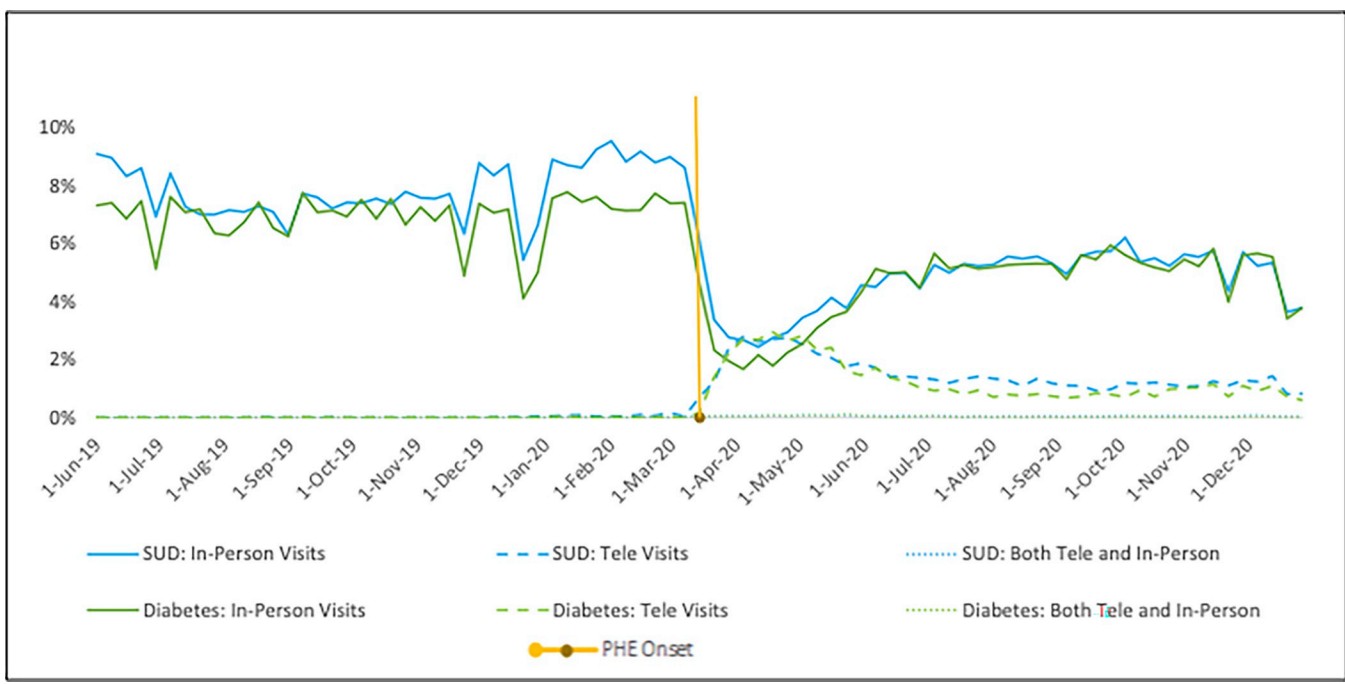

**Fig 1. Percentage of continuously enrolled Wisconsin Medicaid beneficiaries with a SUD or diabetes diagnosis with visits in the week before and after the PHE[a,b].** Abbreviations: SUD, substance use disorder; Tele, telehealth; IP, in-person; PHE, public health emergency. [a]Visits in the week include a primary care visit for any diagnosis (not restricted to visits for substance use disorders or diabetes).[b]A small percentage of cohort members completed both an in-person and telehealth visit in the week, represented by the dotted lines.

exhibited a comorbid psychotic disorder (15.46% vs 5.32%; p = 0.026). Beneficiaries with diabetes were more often Hispanic (11.94% vs 6.63%; p<0.001). Among patients with SUDs, the most prevalent SUDs were opioid use disorder (30.5%), alcohol use disorder (27.2%), cannabis use disorder (8.6%) and stimulant use disorder (4.7%), with 25.4% exhibiting multiple substance use disorders. During the post-PHE period, on average, 1.48% of the SUD cohort and 1.34% of the T2DM cohort had a primary care telehealth visit in the week.

Fig 1 presents the weekly trends in primary care visit utilization for any diagnosis. For both cohorts, the percentage of the cohort with a visit in the week decreased sharply post-PHE. Near zero individuals completed primary care telehealth visits for either cohort pre-PHE followed by a sharp rise and slow decline post-PHE. Beneficiaries with SUDs experienced a larger proportionate drop in primary care utilization for any diagnosis relative to beneficiaries with diabetes. While both cohorts exhibited partial recovery to pre-PHE levels, beneficiaries with diabetes experienced greater recovery than those with SUDs. Of cohort members completing visits in the week post-PHE, a slightly greater percentage of the diabetes cohort initially completed visits via telehealth relative to the SUD cohort. However, over time, the SUD cohort maintained a higher proportionate level relative to the diabetes cohort. A similar pattern persists when restricting visits to those for a concordant disease specific diagnosis, with a slight reduction in the percentage of the diabetes cohort completing an in-person visit in the week (Fig 2).

Table 2 presents results from logistic regression (Models 1 and 2) and fractional regression (Model 3) shown as predicted risk and marginal risk differences (MRD). In Model 1, beneficiaries with SUDs saw a larger decline in primary care utilization post-PHE relative to beneficiaries with diabetes by 0.6 percentage points (MRD: -0.006; 95%CI, -0.008 to -0.003; p<0.001). These results translate into six fewer visits per week per 1000 beneficiaries. Model 2

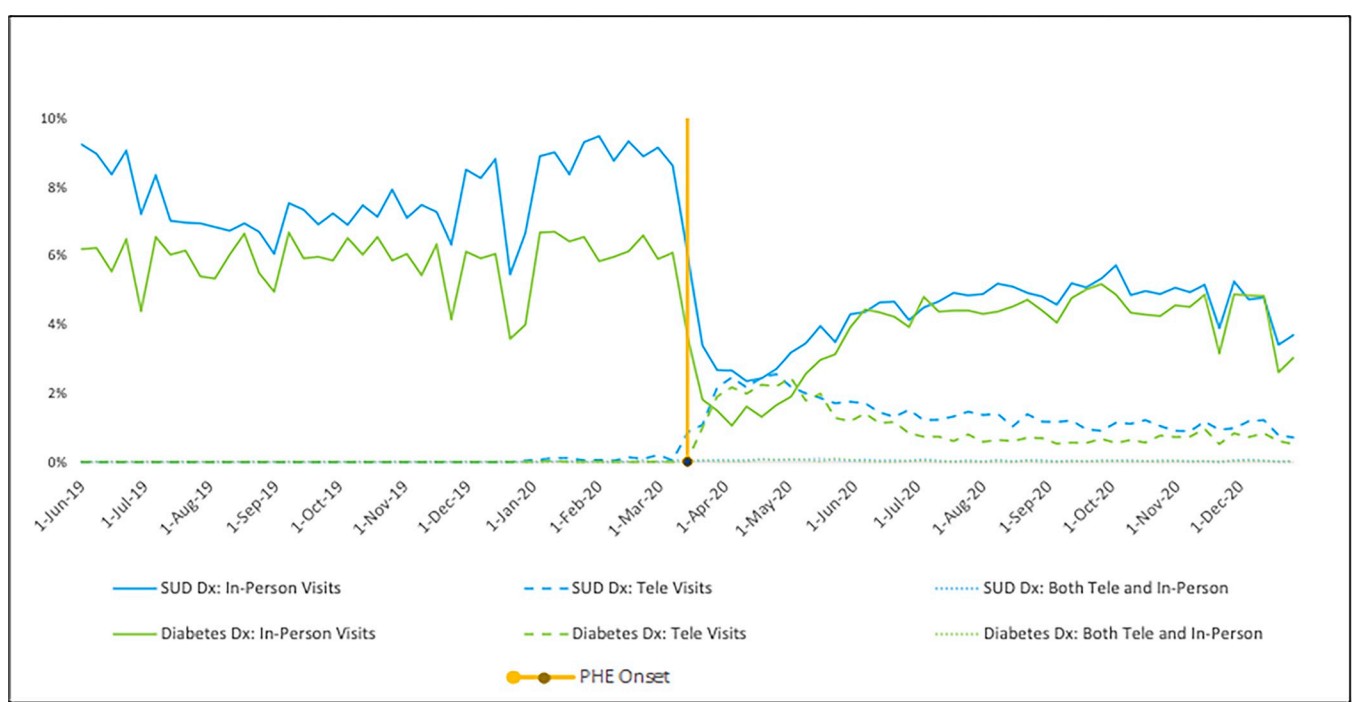

**Fig 2. Percentage of continuously enrolled Wisconsin Medicaid beneficiaries with and SUD or diabetes diagnosis with visits in the week for either a SUD or diabetes diagnosis before and after the PHE[a,b].** Abbreviations: SUD, substance use disorder; Dx, diagnosis; Tele, telehealth; IP, in-person; PHE, public health emergency. [a]Visits in the week include a primary care visit specifically for a substance use disorder or diabetes diagnosis. [b]A small percentage of cohort members completed both an in-person and telehealth visit in the week, represented by the dotted lines.

shows that beneficiaries with SUDs were more likely to use telehealth in the post-PHE period relative to beneficiaries with diabetes by 0.2 percentage points (MRD: 0.002; 95% CI, (0.001 to 0.003); p<0.001), specifically, 2 more visits per week per 1,000 beneficiaries. They also exhibited greater fractional use of telehealth relative to beneficiaries with diabetes. Specifically, the

**Table 2. Predicted risk and marginal risk differences in primary care utilization for any visit, a telehealth visit, and fractional use of telehealth among Wisconsin Medicaid beneficiaries with either a substance use disorder or diabetes.**

| Indicators | 1. Primary care utilization | | | 2. Telehealth utilization | | | 3. Fraction of telehealth utilization | | |
|---|---|---|---|---|---|---|---|---|---|
| | Predicted Risk | SE | P | Predicted Risk | SE | P | Predicted Risk | SE | P |
| | (95% CI) | | | (95% CI) | | | (95% CI) | | |
| Pre-PHE* | 0.069 | 0.001 | <0.001 | 0.00004 | 0 | 0.002 | | | |
| Diabetes | (0.068 to 0.071) | | | (0.00002 to 0.00007) | | | | | |
| Pre-PHE* | 0.08 | 0.001 | <0.001 | 0.0002 | 0 | <0.001 | | | |
| SUD | (0.078 to 0.082) | | | (0.0001 to 0.0003) | | | | | |
| Post-PHE* | 0.058 | 0.001 | <0.001 | 0.013 | 0 | <0.001 | 0.221 | | |
| Diabetes | (0.056 to 0.059) | | | (0.0122 to 0.014) | | | (0.211 to 0.232) | 0.005 | <0.001 |
| Post-PHE* | 0.063 | 0.001 | <0.001 | 0.015 | 0 | <0.001 | 0.24 | | |
| SUD | (0.061 to 0.064) | | | (0.015 to 0.016) | | | (0.230 to 0.250) | 0.005 | <0.001 |
| SUD MRD, Post-PHE | -0.006 | 0.001 | <0.001 | 0.002 | 0.001 | <0.001 | 0.019 | 0.008 | 0.025 |
| | (-0.008 to -0.003) | | | (0.001 to 0.003) | | | (0.004 to .035) | | |
| N | 1,640,329 | | | 1,640,329 | | | 50,411 | | |

Abbreviations: SUD, substance use disorders; SE, standard error; PHE, public health emergency; MRD, marginal risk difference

fraction of telehealth visits for beneficiaries with SUDs was 1.9 percentage points higher than beneficiaries with diabetes (Model 3: MRD: 0.019; 95%CI, 0.004 to 0.035; p = 0.025). In other words, any given visit post-PHE was more likely to be telehealth for beneficiaries with SUDs relative to their diabetic peers.

## Discussion

In a sample of continuously enrolled Wisconsin Medicaid beneficiaries with SUDs or diabetes, we found that patients with SUDs were more likely to use telehealth than patients with diabetes during a period of rapid telehealth expansion at the beginning of the COVID-19 PHE. While patients with diabetes completed a greater proportion of visits via telehealth initially, over time, rates of telehealth utilization remained proportionately higher for patients with SUDs. In the background of this telehealth expansion, we found that patients with SUDs experienced a larger proportionate drop and slower recovery of primary care utilization post-PHE relative to patients with diabetes. Given the disruption observed in SUD primary care utilization, these findings suggest that telehealth played an outsized role in buoying primary care utilization among patients with SUDs compared with patients with diabetes. These findings parallel literature suggesting that remote care models offset in-person declines during the PHE to allow stable access to treatment for opioid use disorder [16].

Similar trends persisted for the SUD cohort when restricting analyses to visits for SUD diagnoses. In contrast, there was only a modest reduction in total visits in the pre- and post-PHE periods for the diabetes cohort when we restricted analyses to visits for a diabetes diagnosis. The clinical implications of these divergent findings are unclear since diagnosis codes do not indicate the content, quantity or quality of treatment. It may be that diseases like diabetes are more easily compartmentalized and thus not addressed at every visit, while SUDs have a more expansive impact on wellness and therefore appear at a higher proportion of visits. Alternatively, it may be that the stigma of substance use makes SUD diagnoses more notable and therefore documented more frequently by clinicians. Given low self-reported rates of SUD treatment receipt [4], the prevalence of SUD diagnosis codes in primary care claims suggests that providers may identify addiction in the office more often than they provide treatment [35].

Our findings carry important health implications for patients with SUDs. Inadequate medical care for addictive and medical disorders results in increased mortality for patients with SUDs [36] alongside preventable medical expenditures [37, 38]. Substantial evidence has demonstrated the health benefits of comprehensive primary care for chronic disease and preventive services [39, 40], including for patients with SUDs [3, 28]. As such, if telehealth expansion increases the likelihood of primary care utilization for patients with SUDs, telehealth may help reduce health disparities while lowering health care costs for these patients. Notably, additional strategies will be needed to bolster treatment engagement among patients with SUDs given the relatively modest, though statistically significant, advantage offered by telehealth for visit completion.

Despite telehealth's potential advantages, we identify several reasons to remain cautious. First, it is unclear whether reducing utilization gaps via synchronous telehealth achieves the same outcomes as in-person visits for SUDs or medical disorders. Growing literature suggests diagnostic and treatment equivalence of in-person and telehealth modalities for mental health [41], and adjunct telehealth services can improve outcomes in SUD care [42, 43]. However, studies directly comparing synchronous telehealth and in-person care for addiction treatment are lacking [42]. Similarly, growing evidence suggests equivalent effectiveness in medical domains like anticoagulation management [41] and acute care [44, 45], but few trials directly

compare synchronous telehealth and in-person care for chronic disease management [46] particularly among patients with SUDs.

Second, the relationship between telehealth and utilization is complex. With respect to addiction treatment, patients with SUDs face treatment barriers such as service availability and affordability, stigma, criminalization, impaired psychosocial functioning, and varying levels of readiness for change [7, 8]. Telehealth can mitigate some barriers by heightening patient anonymity, reducing the need for transportation, enhancing scheduling flexibility, and decreasing the time required to receive care, all of which lower the threshold for treatment engagement [9]. These features may be particularly relevant for patients with limited financial resources, which is common among patients with SUDs [11, 12], who often have less job flexibility [47] and unreliable transportation [10].

On the other hand, telehealth requires digital literacy, access to technology, internet, and a location to receive remote health care [17]. Among patients with SUDs, high rates of homelessness and financial insecurity create barriers to telehealth [12, 17, 18]. The failure of telehealth to fully compensate for PHE primary care disruptions may indicate barriers to telehealth in the Wisconsin Medicaid population. Critically, disproportionate rates of poverty, homelessness and justice-involvement among patients with SUDs from racial and ethnic minority groups demonstrate how structural racism continues to limit treatment utilization [48, 49]. Without intentional deployment of telehealth services, telehealth expansion could unintentionally exacerbate racial disparities by further facilitating access for those with increased financial and social resources [49]. Health systems will need to be attentive to the downstream effects of telehealth expansion to ensure that such initiatives decrease, rather than deepen, health disparities for patients with SUDs.

Finally, telehealth benefits may depend on clinical characteristics like SUD type. For example, treatment for opioid use disorder emphasizes medications while stimulant use disorder emphasizes behavioral interventions [50]. While some evidence suggests that telehealth represents an effective platform for medication management of opioid use disorder [51], the same may not be true for complex behavioral interventions like contingency management for stimulant use disorder. As a result, telehealth may be more appropriate for certain SUDs. Alternatively, telehealth benefits may vary by disease severity. Research demonstrates higher rates of telehealth services among those with more severe SUDs [52]. If these trends persist, telehealth might expand treatment for severe SUDs, but do less to prevent disease progression.

This study had several limitations. First, we required continuous enrollment to minimize bias from compositional changes. This restriction may limit the generalizability of findings to the broader Medicaid population, which often exhibits churning [53]. The MOE protections preventing Medicaid disenrollment during the PHE lessens this limitation. Second, we required an established SUD or diabetes diagnosis prior to study start. Thus, our findings do not reflect utilization among newly diagnosed, undiagnosed, or previously diagnosed individuals without a claim in the six months before study start. Relatedly, use of ICD10 codes from claims to identify individuals with diabetes and SUDs has limitations. As a result, we may have misclassified individuals by type of diabetes or presence of an SUD [54, 55]. Third, in using claims data, we may have missed disease-specific services if SUD or diabetes diagnoses were not recorded. Fourth, we may have missed telehealth services that were miscoded or not submitted for reimbursement. Fifth, it is likely that the patients in the SUD and diabetes cohorts differ in ways not captured in the data, such as presence of carceral involvement and housing instability. These factors also generate barriers to in person care. As a result, our findings may underestimate the engagement advantage of expanding telehealth services. Finally, our analyses may not be generalizable to other states.

In sum, due to the considerable adverse health effects of substance use, patients with SUDs experience disproportionate morbidity and mortality. Yet, few patients with SUDs receive

adequate treatment for medical and addiction diagnoses. In this study, we found that telehealth expansion increased treatment utilization for patients with SUDs beyond that observed for patients with diabetes. This advantage helped mitigate the greater disruption in health care experienced by patients with SUDs due to the PHE. Notably, telehealth expansion was unable to fully compensate for these disruptions in care for either patients with SUDs or diabetes. The degree to which this deficit in utilization reflects structural barriers to telehealth in low-income patient populations is unclear. Health systems and policymakers attempting to expand telehealth services should create mechanisms to reduce telehealth barriers and track telehealth utilization to ensure that telehealth expansion does not exacerbate disparities in health status or health care utilization for patients with SUDs relative to other patient populations.

## Supporting information

**S1 Appendix. Sample construction.**
(DOCX)

**S2 Appendix. Comparison of analytic sample to population.**
(DOCX)

**S3 Appendix. Definition of outcome measures.**
(DOCX)

**S4 Appendix. Regression models technical appendix.**
(DOCX)

## Acknowledgments

The authors of this work are solely responsible for the content therein. The authors thank the Wisconsin Department of Health Services for the use of data for this analysis, but the agency does not certify the accuracy of the analyses presented. This work is done in affiliation and partnership with the University of Wisconsin Institute for Research on Poverty.

## Author Contributions

**Conceptualization:** Alyssa Shell Tilhou, Marguerite Burns.

**Data curation:** Alyssa Shell Tilhou, Laura Dague, Preeti Chachlani, Marguerite Burns.

**Formal analysis:** Alyssa Shell Tilhou, Laura Dague, Preeti Chachlani, Marguerite Burns.

**Funding acquisition:** Laura Dague, Marguerite Burns.

**Methodology:** Alyssa Shell Tilhou, Laura Dague, Marguerite Burns.

**Visualization:** Alyssa Shell Tilhou, Laura Dague.

**Writing – original draft:** Alyssa Shell Tilhou.

**Writing – review & editing:** Alyssa Shell Tilhou, Laura Dague, Preeti Chachlani, Marguerite Burns.

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
