## [Decision Letter · Decision Letter 0]

12 Jul 2023

PONE-D-23-11600Can telehealth expansion boost health care utilization specifically for patients with substance use disorders relative to patients with other types of chronic disease?PLOS ONE

Dear Dr. Tilhou,

Thank you for submitting your manuscript to PLOS ONE. After careful consideration, we feel that it has merit but does not fully meet PLOS ONE’s publication criteria as it currently stands. Therefore, we invite you to submit a revised version of the manuscript that addresses the points raised during the review process.

We look forward to receiving your revised manuscript.

Kind regards,

Meghana Ray, Ph.D., MBA, B.Pharm

Academic Editor

PLOS ONE

Journal Requirements:

Reviewers' comments:

Reviewer's Responses to Questions

**Comments to the Author**

1. Is the manuscript technically sound, and do the data support the conclusions?

Reviewer #1: Yes

Reviewer #2: Yes

2. Has the statistical analysis been performed appropriately and rigorously? 

Reviewer #1: Yes

Reviewer #2: Yes

3. Have the authors made all data underlying the findings in their manuscript fully available?

Reviewer #1: No

Reviewer #2: Yes

4. Is the manuscript presented in an intelligible fashion and written in standard English?

Reviewer #1: Yes

Reviewer #2: Yes

5. Review Comments to the Author

Reviewer #1: This study compared the impact of telehealth expansion during the COVID-19 public health emergency on primary care utilization for patients with substance use disorders and diabetes. This paper fills an important gap in the literature by conducting a comparative analysis focused on low-income patient populations on Medicaid.

While I appreciate the authors efforts to choose an appropriate comparison group, Medicaid beneficiaries with diabetes differ substantially from beneficiaries with SUD. This is demonstrated in the table, in which all sociodemographic variables are statistically significantly different between groups despite exhibiting similar insurance coverage. The biggest limitation of this analysis is that there are likely several other characteristics that differ between the populations (e.g., criminal-legal involvement, stigma, employment, homelessness, etc.) and impact health care utilization which were not accounted for.

The authors highlight that this study cohort includes nonpregnant, nondisabled adults. It is important to highlight that this cohort also represents noninstitutionalized adults.

It would be helpful if the authors could provide a breakdown of the SUDs to understand which SUDs are most/least prevalent. This has important implications for interpreting the findings because treatment and care is vastly different across SUDs.

While all analyses resulted in statistically significant findings, the differences were often small (e.g., 0.5 percentage point difference in Model 1, which translates to a difference of 5 visits per 1,000 beneficiaries). The discussion section should explain clinical significance – is 5 office visits per 1,000 persons clinically meaningful?

Lines 265-266: There is an important typo (you report a comparison between beneficiaries with diabetes and beneficiaries with diabetes).

Lines 351-352: How would you have misclassified individuals by type of diabetes?

Reviewer #2: In PONE-D-23-11600, “Can telehealth expansion boost health care utilization specifically for patients with substance use disorders relative to patients with other types of chronic disease?” the authors use administrative data from Wisconsin’s Medicaid program to examine patterns in primary care telehealth use for beneficiaries with SUDs relative to beneficiaries with Type 2 Diabetes. The authors find that at the onset of the public health emergency, beneficiaries with SUDs were less likely to have primary care encounters than beneficiaries with diabetes, but more likely to have telehealth encounters. This presents important evidence about the take-up of telehealth among Medicaid beneficiaries with SUDs that can inform efforts to increase access to telehealth for this group.

I have a few comments:

1. My primary comment is that PlosOne is not a medical journal so understanding the role of primary care in treatment of patients with diagnoses of SUD and Type 2 Diabetes is critical for the reader. How much of the treatment for these two conditions is in the primary care setting? What do providers do in these encounters that they are able to do remotely? (How diabetes can be managed remotely is explained briefly for diabetes lines 108-109 but is not explained for SUD. Additional context for diabetes would be helpful as well as providing context for telehealth encounters to treat SUD in the primary care setting.)

2. The authors include comorbid psychotic disorder as a covariate and it would be helpful to control other comorbidities. The presence of other chronic conditions would likely be correlated with use of telehealth. As mentioned in the introduction, patients who are also managing cardiac, pulmonary, renal, or hepatic disease might be more likely to have in-person visits because management of those conditions benefits from in-person encounters and/or laboratory work.

3. In the conclusion, the authors discuss that telehealth may be more appropriate for some types of SUDs (though of the two examples it is not stated which telehealth is more appropriate for—opioid use disorder or stimulant use disorder). Is it possible to examine whether the relative increase in telehealth you see is widely experienced across all types of SUD or more prevalent for some types of SUD?

4. In the conclusion, the authors note that those with more severe SUDs have higher rates of telehealth use. Can the authors identify severe SUD in their data and confirm whether the relative increase you observe is widely experienced among all levels of SUD or concentrated among severe SUD?

5. A key sample restriction is that the beneficiaries must be continuously enrolled in Medicaid 6/19-12/20. This results in losing nearly 70 percent of the sample. Is this in line with other work in the pre-COVID era that makes a similar restriction?

6. To understand who is dropped by the continuous enrollment restriction, it would be helpful to add two columns to Appendix Table 2 showing means and % for observable characteristics for respondents with at least one month of parent/caretaker or childless adult eligibility but no continuous enrollment June 2019-Dec 2020.

7. In introducing Figure 2, the authors note that beneficiaries with diabetes are more likely to have a primary care visit that does not note a diabetes diagnosis. If possible, it would be helpful to provide some context for why primary care providers are less likely to list diabetes as a diagnosis code than SUD.

8. Line 265: I think the text should read “Specifically, the fraction of telehealth visits for beneficiaries with SUDs was 1.8 percentage points higher than beneficiaries with diabetes (Model 3: MRD: 0.018; 95% …)”

9. Line 138: I think you need to add “diagnosis” after diabetes

6. PLOS authors have the option to publish the peer review history of their article (what does this mean?). If published, this will include your full peer review and any attached files.

Reviewer #1: No

Reviewer #2: No

---

## [Author Response · Author response to Decision Letter 0]

18 Sep 2023

Dear Dr. Ray and the Editorial Board of PLOS ONE,

Thank you for your recent feedback on our submitted manuscript, “Can telehealth expansion boost health care utilization specifically for patients with substance use disorders relative to patients with other types of chronic disease?” We are pleased to submit a revised version. As requested, we have provided a point-by-point response to both reviewers’ comments and completed the necessary formatting and file adjustments. Thank you in advance for considering the revised version of our manuscript. We feel the article is improved through this revision process. 

Sincerely,

Alyssa Shell Tilhou, MD, PhD

Boston University Medical Center

Response to Reviewers 

Reviewer #1: 

This study compared the impact of telehealth expansion during the COVID-19 public health emergency on primary care utilization for patients with substance use disorders and diabetes. This paper fills an important gap in the literature by conducting a comparative analysis focused on low-income patient populations on Medicaid.

While I appreciate the authors efforts to choose an appropriate comparison group, Medicaid beneficiaries with diabetes differ substantially from beneficiaries with SUD. This is demonstrated in the table, in which all sociodemographic variables are statistically significantly different between groups despite exhibiting similar insurance coverage. The biggest limitation of this analysis is that there are likely several other characteristics that differ between the populations (e.g., criminal-legal involvement, stigma, employment, homelessness, etc.) and impact health care utilization which were not accounted for.

We appreciate the relevance of this comment and have added our inability to control for these and other social factors influencing health care utilization to the limitations section (lines 381-384). Of note, factors like criminal-legal involvement, stigma and homelessness increase challenges to in-person care, specifically, that may be somewhat mitigated by transition to remote telehealth care. As a result, while these differences are a limitation to the study, our findings likely underestimate the advantage of telehealth expansion for patients with SUDs relative to patients with diabetes.

The authors highlight that this study cohort includes nonpregnant, nondisabled adults. It is important to highlight that this cohort also represents noninstitutionalized adults.

This cohort characteristic has been added to the methods section (line 151). 

It would be helpful if the authors could provide a breakdown of the SUDs to understand which SUDs are most/least prevalent. This has important implications for interpreting the findings because treatment and care is vastly different across SUDs.

In this sample, roughly 30.5% have OUD, 27.2% have alcohol use disorder, 8.6% have cannabis use disorder, 4.7% have a stimulant use disorder, 0.4% have a sedative use disorder, 3.5% have a different type of substance use disorder and 25.2% have two or more substance use disorders. We have included this information in the text of the results section (line 226-229).

While all analyses resulted in statistically significant findings, the differences were often small (e.g., 0.5 percentage point difference in Model 1, which translates to a difference of 5 visits per 1,000 beneficiaries). The discussion section should explain clinical significance – is 5 office visits per 1,000 persons clinically meaningful?

Thank you for this comment. To clarify, these are visits per week. In the example provided, 5 office visits per week would translate into 260 per year (5 x 52 weeks), or 26 visits per 100 beneficiaries. We have added this clarification to the results (lines 275 and 278) and a caveat to the discussion section in lines 326-328, shown below.

“Notably, additional strategies will be needed to bolster treatment engagement among patients with SUDs given the relatively modest, though statistically significant, advantage offered by telehealth for visit completion.”

Lines 265-266: There is an important typo (you report a comparison between beneficiaries with diabetes and beneficiaries with diabetes).

Thank you for this correction. This error has been corrected (now line 280). 

Lines 351-352: How would you have misclassified individuals by type of diabetes?

There are a large number of ICD10 diabetes codes, many of which are not type 1 diabetes but also do not indicate type 2 diabetes, our target diagnostic group. There is documented user error assigning the wrong diagnostic code based on type of diabetes diagnosis.1 It is also possible to misclassify patients with substance use disorder based on ICD10 codes.2 We have added the possibility for misclassification of individuals with SUDs, as well as diabetes, to the manuscript: 

“Relatedly, use of ICD10 codes from claims to identify individuals with diabetes and SUDs has limitations. As a result, we may have misclassified individuals by type of diabetes or presence of an SUD.”

Reviewer #2: 

In PONE-D-23-11600, “Can telehealth expansion boost health care utilization specifically for patients with substance use disorders relative to patients with other types of chronic disease?” the authors use administrative data from Wisconsin’s Medicaid program to examine patterns in primary care telehealth use for beneficiaries with SUDs relative to beneficiaries with Type 2 Diabetes. The authors find that at the onset of the public health emergency, beneficiaries with SUDs were less likely to have primary care encounters than beneficiaries with diabetes, but more likely to have telehealth encounters. This presents important evidence about the take-up of telehealth among Medicaid beneficiaries with SUDs that can inform efforts to increase access to telehealth for this group.

I have a few comments:

1. My primary comment is that PlosOne is not a medical journal so understanding the role of primary care in treatment of patients with diagnoses of SUD and Type 2 Diabetes is critical for the reader. How much of the treatment for these two conditions is in the primary care setting? What do providers do in these encounters that they are able to do remotely? (How diabetes can be managed remotely is explained briefly for diabetes lines 108-109 but is not explained for SUD. Additional context for diabetes would be helpful as well as providing context for telehealth encounters to treat SUD in the primary care setting.)

Thank you for this recommendation. We have provided increased context describing 1) the role of primary care for both conditions (lines 130-133) and 2) the effectiveness and content of telehealth for SUD care (lines 98-101) and diabetes care (lines 113-117). 

“Given the central role of primary care in providing addiction and medical services to patients with SUDs and diabetes,3–6 examining the impact of telehealth incorporation on primary care utilization holds particular relevance for reducing utilization disparities for patients with SUDs.”

“Telehealth may also represent an effective modality for treating patients with SUDs given mounting evidence that the core of SUD care – counseling and medication management – can be delivered via telehealth with at or near comparable outcomes to in person care.”7–10

“Specifically, patients with type 2 diabetes offer a compelling comparison group. In contrast with management of cardiac, pulmonary, renal and hepatic disease, which is more substantially improved by in-person examination and laboratory data, much of the management of type 2 diabetes can occur remotely via collaborative retrospective evaluation of home glucose monitoring.”

2. The authors include comorbid psychotic disorder as a covariate and it would be helpful to control other comorbidities. The presence of other chronic conditions would likely be correlated with use of telehealth. As mentioned in the introduction, patients who are also managing cardiac, pulmonary, renal, or hepatic disease might be more likely to have in-person visits because management of those conditions benefits from in-person encounters and/or laboratory work.

Thank you for this suggestion. Incorporating a chronic disease covariate, our models now control for the presence of chronic disease comorbidities using a binary factor representing the baseline presence of one of the following chronic conditions: asthma, chronic kidney disease, chronic obstructive pulmonary disease, coronary artery disease, hypertension, thyroid disorders, heart failure, chronic liver disease, and osteoarthritis. This variable is described in lines 189-191. 

“Chronic medical conditions included asthma, chronic kidney disease, chronic obstructive pulmonary disease, coronary artery disease, hypertension, thyroid disorders, heart failure, chronic liver disease, and osteoarthritis.”

Our models in the revised version of the manuscript and appendix reflect updated regression results including this new covariate. The addition of a chronic disease covariate did not substantively impact results. Patients with SUDs still exhibited a larger decline in primary care utilization post-PHE relative to those with diabetes (by 0.6 percentage points instead of 0.5 percentage points with a p value <0.001). Beneficiaries with SUDs exhibited greater likelihood of telehealth utilization at the same margin as without the chronic disease covariate (0.2 percentage points, p<0.001). The difference in fraction of visits completed by telehealth also remained similar: beneficiaries with SUDs exhibited a greater fraction of telehealth visits relative to beneficiaries with diabetes at a margin of 0.019 instead of 0.018 (p=0.025). These results can be seen in Table 2 of the manuscript with the technical results now shown in Appendix 4.

3. In the conclusion, the authors discuss that telehealth may be more appropriate for some types of SUDs (though of the two examples it is not stated which telehealth is more appropriate for—opioid use disorder or stimulant use disorder). Is it possible to examine whether the relative increase in telehealth you see is widely experienced across all types of SUD or more prevalent for some types of SUD?

In a parallel paper, we have focused on trends in telehealth expansion by SUD type. We hope this paper will be published in the near future. In the meantime, we have clarified the different advantages and limitations for the two referenced SUDs in lines 362-365.

“While some evidence suggests that telehealth represents an effective platform for medication management of opioid use disorder, the same may not be true for complex behavioral interventions like contingency management for stimulant use disorder.”

4. In the conclusion, the authors note that those with more severe SUDs have higher rates of telehealth use. Can the authors identify severe SUD in their data and confirm whether the relative increase you observe is widely experienced among all levels of SUD or concentrated among severe SUD?

We are not able to assess severity in the claims data. Rather, in referencing previously published work, we intended to identify the possibility that telehealth expansion may not equally support all patients with SUDs and may be more preferentially used by those with increased disease severity. 

5. A key sample restriction is that the beneficiaries must be continuously enrolled in Medicaid 6/19-12/20. This results in losing nearly 70 percent of the sample. Is this in line with other work in the pre-COVID era that makes a similar restriction?

Thank you for this question, which drew our attention to an error in Appendix 1, our sample construction table. Here we report the population with at least one month of eligibility June 2019-December 2020. However, the appropriate initial population to display are adults enrolled in June 2019 via parent/caretaker or childless adult eligibility and no eligibility due to pregnancy at any point June 2019-December 2020 (N=273,105). We have revised the original table, which is pasted below and included in a tracked and corrected version of the appendix (rows 1-4). In addition, we repeated the sample construction process one year prior to our study population based on enrollment in June 2018 (rows 5-8) to allow comparison with a time frame fully preceding the pandemic (N=277,591). The results from this comparison cohort are included below. In the study cohort, 52.7% of the initial population are continuously enrolled for 18 months (N=143,992); in the pre-PHE cohort, 40.5% of the initial population are continuously enrolled for 18 months (n=112,286). The increased population with continuous enrollment in the study cohort relative to the pre-PHE cohort is likely due to the maintenance of eligibility protections provided by the Families First Coronavirus Response Act, which prohibited Medicaid beneficiary disenrollment during the PHE (and therefore resulted in an increased proportion of beneficiaries with continuous enrollment). This means that our cohort is more representative of the general population than it might typically be, due to the lower attrition rate.

Appendix 1. Sample Construction

Inclusion/Exclusion Criteria for Study Population Unique Individuals

(1) Population: Individuals ages 18-64 enrolled in June 2019 via parent/caretaker or childless adult eligibility 284,421

(2) Without any eligibility June 2019-December 2020 due to pregnancy 273,105

(3) Continuous enrollment from June 2019 – December 2020 143,992

(4) Diagnosis of SUD in any position on outpatient, inpatient, or emergency department claim, December 2018-May 2019; OR Diagnosis of type 2 diabetes in any position on outpatient, inpatient, or emergency department claim, December 2018-May 2019 SUD 17,336

Diabetes 8,499

Inclusion/Exclusion Criteria for Comparison Population Unique Individuals

(5) Population: Individuals ages 18-64 enrolled in June 2018 via parent/caretaker or childless adult eligibility 290,696

(6) Without any eligibility June 2018-December 2019 due to pregnancy 277,591

(7) Continuous enrollment from June 2018 – December 2019 112,286

(8) Diagnosis of SUD in any position on outpatient, inpatient, or emergency department claim, December 2017-May 2018; OR Diagnosis of type 2 diabetes in any position on outpatient, inpatient, or emergency department claim, December 2017-May 2018 SUD 13,546

Diabetes 7,235

Abbreviations: SUD, substance use disorder

6. To understand who is dropped by the continuous enrollment restriction, it would be helpful to add two columns to Appendix Table 2 showing means and % for observable characteristics for respondents with at least one month of parent/caretaker or childless adult eligibility but no continuous enrollment June 2019-Dec 2020.

Thank you for this suggestion. We have completed this request in Appendix 2. We begin with the population enrolled in June 2019 without any eligibility in the study period due to pregnancy (N=273,105). We then remove N=143,992 individuals with continuous enrollment in the study period leaving N=129,113 who were not enrolled continuously. The characteristics of these individuals are now shown in the first two columns of Appendix 2, also included below. Individuals without continuous enrollment exhibit a higher proportion parents/caretakers and psychotic disorder, and lower proportion missing geography and with income ≤50% federal poverty level. Otherwise, the two groups are fairly similar.

Appendix 2. Comparison of analytic sample to population of Wisconsin Medicaid beneficiaries enrolled June 2019 and with continuous enrollment 06/01/2019–12/31/2020 via Parent/Caretaker or Childless Adult eligibility 

 Not continuously Enrolled Continuously Enrolled SUD Cohort Diabetes Cohort

N=unique subjects 129,113 100.00% 143,992 100.00% 17,336 100.00% 8,499 100.00%

Eligibility Category 

 Childless Adults 66,741 51.69% 82,434 57.25% 11,841 68.31% 5,373 63.22%

 Parents/Caretakers 62,372 48.31% 61,558 42.75% 5,495 31.69% 3,126 36.78%

Comorbid diagnoses 

 Psychotic disorder 14,319 11.09% 6,778 4.71% 2,680 15.46% 452 5.32%

 SUD and Diabetes 2,200 1.70% 1,111 0.77% 1,111 6.41% 0 0.00%

 Chronic disease 25,331 19.62% 31,671 21.99% 5,845 33.72% 5748 67.63%

Sex 

 Female 71,755 55.58% 81,094 56.31% 8,301 47.88% 4,703 55.34%

 Male 57,358 44.42% 62,898 43.69% 9,035 52.12% 3,796 44.66%

 Missing 0 0.00% 0 0.00% 0 0.00% 0 0.00%

Age 

 Mean age (SD) 37.24 (12.12) 39 (11.89) 38.88(10.34) 48(9.88)

Race 

 American Indian 2,985 2.31% 3,332 2.32% 799 4.62% 300 3.53%

 Asian 4,097 3.17% 4,687 3.25% 107 0.62% 424 4.99%

 Black 29,331 22.72% 29,082 20.19% 2,525 14.54% 1,581 18.60%

 Multiracial 3,257 2.52% 2,880 1.97% 368 2.09% 126 1.48%

 Pacific Islander 171 0.13% 211 0.15% 15 0.09% 23 0.27%

 White 76,769 59.46% 90,397 62.70% 12,340 71.13% 4,880 57.42%

 Race Missing 12,503 9.68% 13,403 9.43% 1,182 6.91% 1,165 13.71%

Ethnicity 

 Hispanic 11,858 9.18% 11,698 8.11% 1,154 6.63% 1,015 11.94%

 Not Hispanic 114,713 88.85% 129,846 90.13% 15,972 92.13% 7,334 86.29%

 Missing 2,542 1.97% 2,448 1.76% 210 1.24% 150 1.76%

Education 

 > High School 73,711 57.09% 80,258 55.69% 10,255 59.12% 4,452 52.38%

< High School 21,801 16.89% 26,464 18.39% 3,440 19.85% 1,558 18.33%

 Missing 33,601 26.02% 37,270 25.92% 3,641 21.03% 2,489 29.29%

Geography 

 Urban 90,479 70.08% 92,310 64.10% 11,223 64.66% 5,288 62.22%

 Rural 29,858 23.13% 31,327 21.77% 3,573 20.59% 1,885 22.18%

 Missing 8,776 6.80% 20,355 14.13% 2,540 14.74% 1,326 15.60%

Income 

 ≤50% FPL 82,077 63.57% 104,860 72.82% 14,560 83.99% 5,659 66.58%

 50-100% FPL 47,036 36.43% 39,132 27.18% 2,776 16.01% 2,840 33.42%

 >100% FPL 0 0.00% 0 0.00% 0 0.00% 0 0.00%

 Missing 0 0.00% 0 0.00% 0 0.00% 0 0.00%

Abbreviations: SUD, substance use disorder; SD, standard deviation; FPL, federal poverty level

7. In introducing Figure 2, the authors note that beneficiaries with diabetes are more likely to have a primary care visit that does not note a diabetes diagnosis. If possible, it would be helpful to provide some context for why primary care providers are less likely to list diabetes as a diagnosis code than SUD.

Thank you, we also considered this difference to be an interesting finding. In lines 311-314, we discuss that diseases like diabetes may be more easily compartmentalized and thus not addressed at every visit. In contrast, SUDs have a more expansive impact on wellness and therefore appear at a higher proportion of visits. We have added one more theory to bolster this aspect of the discussion (line 362-365). 

“It may also be that the stigma of substance use makes SUD diagnoses more notable and therefore documented more frequently by clinicians.”

8. Line 265: I think the text should read “Specifically, the fraction of telehealth visits for beneficiaries with SUDs was 1.8 percentage points higher than beneficiaries with diabetes (Model 3: MRD: 0.018; 95% …)”

 Thank you very much. This has been corrected. 

9. Line 138: I think you need to add “diagnosis” after diabetes

 Thank you very much. This has been corrected.

References

1. Rashidian S, Abell-Hart K, Hajagos J, et al. Detecting miscoded diabetes diagnosis codes in electronic health records for quality improvement: temporal deep learning approach. JMIR Med Inform. 2020;8(12):e22649.

2. Lagisetty P, Garpestad C, Larkin A, et al. Identifying individuals with opioid use disorder: Validity of International Classification of Diseases diagnostic codes for opioid use, dependence and abuse. Drug Alcohol Depend. 2021;221:108583. doi:10.1016/j.drugalcdep.2021.108583

3. LaBelle CT, Han SC, Bergeron A, Samet JH. Office-based opioid treatment with buprenorphine (OBOT-B): statewide implementation of the Massachusetts collaborative care model in community health centers. J Subst Abuse Treat. 2016;60:6-13. doi:10.1016/j.jsat.2015.06.010

4. Cheng HY, McGuinness LA, Elbers RG, et al. Treatment interventions to maintain abstinence from alcohol in primary care: systematic review and network meta-analysis. BMJ. 2020;371.

5. Davidson JA. The increasing role of primary care physicians in caring for patients with type 2 diabetes mellitus. In: Mayo Clinic Proceedings. Vol 85. Elsevier; 2010:S3-S4.

6. Olfson M, Zhang V, Schoenbaum M, King M. Buprenorphine Treatment By Primary Care Providers, Psychiatrists, Addiction Specialists, And Others: Trends in buprenorphine treatment by prescriber specialty-primary care providers, psychiatrists, and addiction medicine specialists. Health Aff (Millwood). 2020;39(6):984-992.

7. Kruse CS, Lee K, Watson JB, Lobo LG, Stoppelmoor AG, Oyibo SE. Measures of effectiveness, efficiency, and quality of telemedicine in the management of alcohol abuse, addiction, and rehabilitation: systematic review. J Med Internet Res. 2020;22(1):e13252.

8. Mark TL, Treiman K, Padwa H, Henretty K, Tzeng J, Gilbert M. Addiction treatment and telehealth: review of efficacy and provider insights during the COVID-19 pandemic. Psychiatr Serv. 2022;73(5):484-491.

9. Mochari-Greenberger H, Pande RL. Behavioral health in America during the COVID-19 pandemic: Meeting increased needs through access to high quality virtual care. Am J Health Promot. 2021;35(2):312-317.

10. Tilhou AS, Dague L, Saloner B, Beemon D, Burns M. Trends in Engagement With Opioid Use Disorder Treatment Among Medicaid Beneficiaries During the COVID-19 Pandemic. JAMA Health Forum. 2022;3(3):e220093-e220093.

---

## [Decision Letter · Decision Letter 1]

26 Dec 2023

PONE-D-23-11600R1Can telehealth expansion boost health care utilization specifically for patients with substance use disorders relative to patients with other types of chronic disease?PLOS ONE

Dear Dr. Tilhou,

The reviewers have made some additional comments regarding formatting and minor changes that need to be addressed prior to reaching a decision on the manuscript. There are no conflicts with the reviewers comments and I look forward to receiving your manuscript with the requested changes.

We look forward to receiving your revised manuscript.

Kind regards,

Meghana Ray, Ph.D.

Academic Editor

PLOS ONE

Journal Requirements:

Reviewers' comments:

Reviewer's Responses to Questions

**Comments to the Author**

1. If the authors have adequately addressed your comments raised in a previous round of review and you feel that this manuscript is now acceptable for publication, you may indicate that here to bypass the “Comments to the Author” section, enter your conflict of interest statement in the “Confidential to Editor” section, and submit your "Accept" recommendation.

Reviewer #1: All comments have been addressed

Reviewer #2: All comments have been addressed

2. Is the manuscript technically sound, and do the data support the conclusions?

Reviewer #1: Yes

Reviewer #2: Yes

3. Has the statistical analysis been performed appropriately and rigorously? 

Reviewer #1: Yes

Reviewer #2: Yes

4. Have the authors made all data underlying the findings in their manuscript fully available?

Reviewer #1: No

Reviewer #2: Yes

5. Is the manuscript presented in an intelligible fashion and written in standard English?

Reviewer #1: Yes

Reviewer #2: Yes

6. Review Comments to the Author

Reviewer #1: (No Response)

Reviewer #2: The revision addresses the points I raised on the initial submission. I have a few formatting/presentation suggestions:

Discussion of Fig 1: I cannot see on the figure that diabetes initially had more visits (may be that the chosen colors make it hard to differentiate?) (around line 212)

Table 1: Didn’t you add other comorbid conditions that would make more sense to show in the Comorbid diagnosis panel other than “SUD and diabetes”?

Table 1: consider removing “missing sex” row? (because it is 0 for all?)

Table 1: consider removing “missing income” row? (because it is not missing for any?)

Figure 1: change label to read “in person” not IP? (Some health econ readers might infer IP=inpatient?)

Titles to Fig 1 and 2: I think you need to add “diagnosis”

Parentheses to close on line 200

Exhibiting not exhibited on line 202

7. PLOS authors have the option to publish the peer review history of their article (what does this mean?). If published, this will include your full peer review and any attached files.

Reviewer #1: No

Reviewer #2: No

---

## [Author Response · Author response to Decision Letter 1]

20 Jan 2024

Dear Dr. Ray and the Editorial Board of PLOS ONE,

Thank you for your recent feedback on our submitted manuscript, “Can telehealth expansion boost health care utilization specifically for patients with substance use disorders relative to patients with other types of chronic disease?” We are pleased to submit a revised version. As requested, we have provided a point-by-point response to both reviewers’ comments and completed the necessary formatting and file adjustments. Thank you in advance for considering the revised version of our manuscript. We feel the article is improved through this revision process. 

Sincerely,

Alyssa Shell Tilhou, MD, PhD

Boston University Medical Center

Response to Reviewers

1. If the authors have adequately addressed your comments raised in a previous round of review and you feel that this manuscript is now acceptable for publication, you may indicate that here to bypass the “Comments to the Author” section, enter your conflict of interest statement in the “Confidential to Editor” section, and submit your "Accept" recommendation.

Reviewer #1: All comments have been addressed

Reviewer #2: All comments have been addressed

2. Is the manuscript technically sound, and do the data support the conclusions?

Reviewer #1: Yes

Reviewer #2: Yes

3. Has the statistical analysis been performed appropriately and rigorously?

Reviewer #1: Yes

Reviewer #2: Yes

4. Have the authors made all data underlying the findings in their manuscript fully available?

Reviewer #1: No

We have included the requested data availability statement. Please let us know if further commentary is needed.

Reviewer #2: Yes

5. Is the manuscript presented in an intelligible fashion and written in standard English?

Reviewer #1: Yes

Reviewer #2: Yes

6. Review Comments to the Author

Reviewer #1: (No Response)

Reviewer #2: The revision addresses the points I raised on the initial submission. I have a few formatting/presentation suggestions:

Discussion of Fig 1: I cannot see on the figure that diabetes initially had more visits (may be that the chosen colors make it hard to differentiate?) (around line 212)

We have corrected the wording in this section to clarify our intent and compare visits across individuals in the diabetes and SUD cohort (line 213 of the revised manuscript).

“Of cohort members completing visits in the week post-PHE, a slightly greater percentage of the diabetes cohort initially completed visits via telehealth relative to the SUD cohort.”

Table 1: Didn’t you add other comorbid conditions that would make more sense to show in the Comorbid diagnosis panel other than “SUD and diabetes”?

Chronic disease comorbidities has been added as a row to Table 1. Thank you for this correction. 

Table 1: consider removing “missing sex” row? (because it is 0 for all?)

Completed (including in the appendix).

Table 1: consider removing “missing income” row? (because it is not missing for any?)

Completed (including in the appendix). 

Figure 1: change label to read “in person” not IP? (Some health econ readers might infer IP=inpatient?)

Completed. 

Titles to Fig 1 and 2: I think you need to add “diagnosis”

Completed. 

Parentheses to close on line 200

Completed. 

Exhibiting not exhibited on line 202

Completed. 

7. PLOS authors have the option to publish the peer review history of their article (what does this mean?). If published, this will include your full peer review and any attached files.

Do you want your identity to be public for this peer review? For information about this choice, including consent withdrawal, please see our Privacy Policy.

Reviewer #1: No

Reviewer #2: No

Configuration confirmed on PACE. 

---

## [Editor Report · Decision Letter 2]

9 Feb 2024

Can telehealth expansion boost health care utilization specifically for patients with substance use disorders relative to patients with other types of chronic disease?

PONE-D-23-11600R2

Dear Dr. Tilhou,

We’re pleased to inform you that your manuscript has been judged scientifically suitable for publication and will be formally accepted for publication once it meets all outstanding technical requirements.

Kind regards,

Meghana Ray, Ph.D., MBA, B.Pharm

Academic Editor

PLOS ONE
---

## [Editor Report · Acceptance letter]

21 Mar 2024

PONE-D-23-11600R2 

PLOS ONE

Dear Dr. Tilhou, 

I'm pleased to inform you that your manuscript has been deemed suitable for publication in PLOS ONE. Congratulations! Your manuscript is now being handed over to our production team.

Kind regards, 

on behalf of

Dr. Meghana Ray 

Academic Editor

PLOS ONE